# Deep Reinforcement Learning-Empowered Cost-Effective Federated Video Surveillance Management Framework

**DOI:** 10.3390/s24072158

**Published:** 2024-03-27

**Authors:** Dilshod Bazarov Ravshan Ugli, Alaelddin F. Y. Mohammed, Taeheum Na, Joohyung Lee

**Affiliations:** 1Department of Computing, Gachon University, Seongnam-si 13120, Republic of Korea; dilshod@gachon.ac.kr; 2Department of International Studies, Dongshin University, 67, Dongshindae-gil, Naju-si 58245, Republic of Korea; alaelddin@dsu.ac.kr; 3Electronics and Telecommunications Research Institute (ETRI), Yuseong-gu, Daejeon 34129, Republic of Korea; taeheum@etri.re.kr

**Keywords:** LSTM, federated learning, DQN, hierarchical edge computing, cost-effective video surveillance management system

## Abstract

Video surveillance systems are integral to bolstering safety and security across multiple settings. With the advent of deep learning (DL), a specialization within machine learning (ML), these systems have been significantly augmented to facilitate DL-based video surveillance services with notable precision. Nevertheless, DL-based video surveillance services, which necessitate the tracking of object movement and motion tracking (e.g., to identify unusual object behaviors), can demand a significant portion of computational and memory resources. This includes utilizing GPU computing power for model inference and allocating GPU memory for model loading. To tackle the computational demands inherent in DL-based video surveillance, this study introduces a novel video surveillance management system designed to optimize operational efficiency. At its core, the system is built on a two-tiered edge computing architecture (i.e., client and server through socket transmission). In this architecture, the primary edge (i.e., client side) handles the initial processing tasks, such as object detection, and is connected via a Universal Serial Bus (USB) cable to the Closed-Circuit Television (CCTV) camera, directly at the source of the video feed. This immediate processing reduces the latency of data transfer by detecting objects in real time. Meanwhile, the secondary edge (i.e., server side) plays a vital role by hosting a dynamically controlling threshold module targeted at releasing DL-based models, reducing needless GPU usage. This module is a novel addition that dynamically adjusts the threshold time value required to release DL models. By dynamically optimizing this threshold, the system can effectively manage GPU usage, ensuring resources are allocated efficiently. Moreover, we utilize federated learning (FL) to streamline the training of a Long Short-Term Memory (LSTM) network for predicting imminent object appearances by amalgamating data from diverse camera sources while ensuring data privacy and optimized resource allocation. Furthermore, in contrast to the static threshold values or moving average techniques used in previous approaches for the controlling threshold module, we employ a Deep Q-Network (DQN) methodology to manage threshold values dynamically. This approach efficiently balances the trade-off between GPU memory conservation and the reloading latency of the DL model, which is enabled by incorporating LSTM-derived predictions as inputs to determine the optimal timing for releasing the DL model. The results highlight the potential of our approach to significantly improve the efficiency and effective usage of computational resources in video surveillance systems, opening the door to enhanced security in various domains.

## 1. Introduction

In recent years, the field of video surveillance has witnessed significant advancements through the application of machine learning (ML) techniques [1]. ML has played a pivotal role in revolutionizing how objects are identified and tracked in video streams, providing enhanced security, safety, and situational awareness in various domains such as public spaces, transportation, and industrial environments. One subset of ML, known as deep learning (DL) [2], has improved video surveillance capabilities by achieving greater accuracy because of its ability to more accurately learn from enormous data patterns using neural networks (NNs) [3], as opposed to ML [4]. DL models have proven to be highly effective at automatically analyzing video data, enabling the detection of objects of interest, and monitoring their movements with unprecedented accuracy [5].

Nevertheless, one of the significant challenges in processing large amounts of video data and running DL models for accurate object identification and tracking in real time lies in the efficient utilization of limited computational resources [6]. DL-based models require a great deal of computing resources, such as (i) GPU computing resources for model inference, and (ii) GPU memory resources for model loading. Specifically, utilizing GPU computing resources for model inference and GPU memory resources for model loading presents a critical bottleneck in video surveillance systems. Despite this, the availability of GPUs is often limited, making efficient resource allocation crucial for maximizing system performance [7,8]. Once a DL model is loaded into GPU memory, it needs to be kept allocated until unloaded. This approach ensures the model is readily available for inference when new frames are processed. Even so, in video surveillance systems, rare events are often captured, resulting in standby GPU memory being wasted for extended periods. This idle memory represents a missed opportunity to utilize GPU resources for other tasks and limits the system’s scalability.

Smart scheduling techniques [9] and resource management frameworks [10] optimize GPU resource usage for video surveillance systems. Smart scheduling dynamically loads and unloads deep learning models on GPUs to match video stream demand, efficiently reclaiming standby resources. Meanwhile, the resource management frameworks automatically allocate GPU resources to different video streams based on availability and demand, adjusting model loading according to system load and priority. This can enhance resource utilization and scalability for surveillance systems operating in complex environments with multiple camera feeds [9,10]. In this context, the AdaMM framework [11] was developed to address the challenges of always loading GPU memory resources for DL-based object movement and motion-tracking models in hierarchical edge computing systems. The main idea behind AdaMM was to introduce a constant threshold value denoted as $\theta_{m}$ to determine when to release the DL model. However, the framework faces two main shortcomings in setting the threshold value $\theta_{m}$. If $\theta_{m}$ is set to a large value, it means that the DL model will be released less frequently. This can lead to increased GPU memory consumption because the DL model remains active for a longer duration. Conversely, if the threshold value $\theta_{m}$ is set too small, it leads to frequent DL model releases and reloads. This constant switching between model states can significantly increase the delay and impose other operational issues. Another framework named CogVSM was proposed in [12], incorporating predictive modeling and smoothing techniques to control the threshold value (i.e., $\theta_{m}$) for releasing the DL model. According to the claim in the CogVSM framework, the Long Short-Term Memory (LSTM) model predicts future object occurrences based on historical data, and these predictions are then passed to smooth the LSTM predictions using the Exponential Weighted Moving Average (EWMA) technique. Based on the smoothed predictions, the threshold value is adjusted dynamically. The authors achieved a significant reduction in GPU memory compared to previous studies. While The CogVSM framework offers a promising approach to dynamic threshold management for DL model release in edge computing systems, it does have certain limitations. Firstly, training a machine learning model like LSTM typically requires access to a substantial amount of data, which is often centralized on a server. As a result, using centralized data for model training can raise privacy concerns, especially when dealing with sensitive information. This centralized approach may not be suitable for applications where data privacy and security are paramount. Also, training a DL (i.e., LSTM) model demands significant computational resources and can be expensive, as mentioned in [13]. Secondly, the limitations of EWMA [14] include its reliance on a static smoothing factor that may not adapt well to diverse scenarios, especially when dealing with non-linear problems or fluctuating object patterns in video surveillance data, which can lead to suboptimal performance in rapidly changing object movement patterns. Additionally, EWMA is a static smoothing technique that lacks learning capability, hindering the ability of the CogVSM framework to continuously optimize threshold values based on evolving data patterns.

In order to address centralized training issues, adopting FL in a novel system can make it safe, private, and computationally efficient to train the LSTM model on massive amounts of decentralized data. FL has emerged as a promising approach to address the challenges associated with centralized data processing and privacy concerns in machine learning applications [15], a concept initially coined by Google [16]. To alleviate the limitations of the EWMA technique, as described in [12], the Deep Q-Network (DQN) model [17] can be used to adapt to changing conditions and optimize the framework’s performance over time. The DQN [17] is a reinforcement learning (RL) technique that combines deep neural networks with the Q-learning algorithm to learn optimal decision-making policies. Correspondingly, this paper investigates the impact of dynamic model release on efficiently utilizing computing resources in video surveillance systems. We explore various strategies for reducing wasted GPU memory usage and propose a novel algorithm that intelligently loads and unloads DL models based on the characteristics of video streams. Through extensive experiments and evaluations, we assess the effectiveness of our proposed method in improving resource utilization, scalability, and real-time performance of video surveillance systems. The contributions of the paper are as follows:Design of a hierarchical edge computing system that uses the You Only Look Once (YOLO) algorithm for object detection in the first-level edge (i.e., client side) and a hierarchical interaction of object occurrence prediction and dynamically controlled threshold modules in the second-level edge (i.e., server side), which helps reduce standby GPU memory and prevent latency during unnecessary model reloading.Instead of relying on fixed threshold values or basic moving averages, our design for the controlling threshold module employs the DQN methodology to dynamically adjust threshold values by utilizing predictions derived from LSTM as inputs to determine the most appropriate timing for releasing the DL model. This effectively strikes a balance between conserving GPU memory and preventing unnecessary reloading of the DL model.Prediction of future object appearance patterns using the LSTM model and controlling the threshold module hierarchically based on the LSTM prediction outcomes, further improving the system’s accuracy.Implementation of federated learning (FL) to train the LSTM model on data from multiple cameras without compromising privacy and efficient use of resources, which addresses the limitations of training on a centralized dataset, including privacy concerns and processing power limitations.Adoption of a Deep Q-Network (DQN) model to make more intelligent decisions about when to trigger the model release based on the object appearance patterns predicted by the FL-based LSTM model, which overcomes the limitations of the fixed smoothing factor and the lack of learning capability of the EWMA technique in the previous work.

The rest of this paper is structured as follows. We discuss the related works in Section 2. Section 3 presents the proposed method. We discuss the framework’s implementation in Section 4. Section 5 describes the experimental setup and analyzes the results obtained from our evaluations. Finally, we conclude with a summary of our findings and provide insights into future research directions in Section 6.

## 2. Related Works

Several approaches have recently been proposed to make DL-based video surveillance with edge computing servers more cost-effective. We have summarized the state of the art (SOTA) in energy-efficient and DL-based video surveillance studies in Table 1 for the convenience of the reader. These approaches, as highlighted in [18,19,20,21,22], have achieved significant improvements in energy efficiency by leveraging edge computing and optimization mechanisms. This has reduced network bandwidth and response time in IoT-based smart video surveillance systems, enabling effective object detection and analysis of abnormal behavior. Ref. [23] presented a specialized architecture centered around edge computing for Unmanned Aerial Vehicle (UAV) settings. It aimed to reduce delays and network data usage by identifying unusual object occurrences. The proposed research emphasized screening video frames of interest on the edge device and transmitting only the frames needing analysis to the cloud server. In [24], the authors introduced an approach that outperformed conventional methods in the precise detection and tracking of objects of interest. They also tackled issues such as minimizing GPU processing requirements and enhancing the accuracy of motion tracking. Several studies [25,26,27,28] on anomaly behavior detection have focused mostly on improving the accuracy of DL models, making them tiny on edge devices. In [29,30,31], the authors analyzed the challenges and potential of DL-based pose anomaly detection in video analysis, emphasizing the advantages in terms of privacy and computational efficiency. Nevertheless, these studies did not consider the hierarchical structure of edge computing systems or the practical considerations related to delivering real-time video surveillance services.

To address these challenges, the AdaMM framework [11] was proposed, which introduced a constant threshold value ($\theta_{m}$) for releasing the DL model (i.e., DL-based object movement and motion-tracking model) in hierarchical edge computing systems. However, the method is insufficient in two aspects. If $\theta_{m}$ is set too large, the frequency of DL model releases decreases, leading to increased GPU memory consumption. Conversely, if the threshold value $\theta_{m}$ is too small, frequent DL model release and reload switching occur, resulting in delays and other issues. Another framework called CogVSM was proposed in [12], in which the authors suggested the use of an LSTM [32] model to predict future object occurrences and employed the EWMA technique to smooth the LSTM prediction results for controlling the threshold value ($\theta_{m}$) to release the DL model. However, the method employed in the CogVSM framework has certain limitations. For instance, training an ML (i.e., LSTM) model using data from a central server raises privacy issues due to potentially sensitive information [33] while also demanding substantial computational resources and costs [13]. Furthermore, the shortcomings of the EWMA method become evident [14], as it relies on a static smoothing factor, which may not be adaptable to diverse scenarios (i.e., non-linear problems), especially when dealing with fluctuating object patterns in video surveillance data, and its lack of learning capability hampers its ability to improve decision making over time through experience.

## 3. Cost-Effective Video Surveillance Management System Model

This section proposes a cost-effective video surveillance management system for hierarchical edge computing systems. The proposed framework, depicted in Figure 1, consists of two edge nodes: the first one handles object detection, whereas the second one manages the prediction of future object occurrences through an FL-based LSTM, a DQN-based controlling threshold, and motion-tracking modules. We assume the first and second edge nodes are connected to each other. Specifically, we focus on detecting and counting people at the first edge for smart DL model release by anticipating object (e.g., people) occurrences in CCTV video frames at the second edge.

Notably, the processes start taking input video frames from the attached Internet Protocol (IP) camera in the first edge node, and then the following four tasks are performed:Task 1: Initiating the object detection process. Upon the arrival of video frames at the first edge, the YOLO object detection algorithm [34] is initiated. Once objects are identified, the first edge node sends the detection information (such as the number of detected people and video frames) to the second edge for further processing.Task 2: Handling object movement and motion tracking. Frames containing the results of the object detection are forwarded to the second edge node. When the second edge node receives both the video frames and detected person counts from the first edge node, they are placed in a processing queue based on their identifications. Subsequently, the detection information is transmitted to the LSTM module for predicting future object occurrences. Task (2) predicts the number of objects expected in future video frames and conveys these predictions to the DQN-based control threshold module. Then, the DQN model receives the predicted data.Task 3: Controlling threshold management. Within this module, an RL-based DQN model simultaneously makes binary decisions to adjust the threshold time value. Ultimately, by employing this threshold, the DQN model within the control threshold module determines whether to issue a stop command or forward the video frames to the motion-tracking module, which is the subsequent task.Task 4: Execution of object movement and motion tracking. If the control module opts to activate a DL-based motion-tracking model, a trigger signal is transmitted, instructing the motion-tracking model to start while also pausing it as needed. Alternatively, video frames are conveyed to the motion-tracking model via a queue. Task (4) is responsible for managing object movement and executing motion tracking whenever video frames are received through the control module.

Our main contribution to the proposed framework is divided into two parts: (i) an FL-based LSTM prediction module, and (ii) a DQN-based controlling threshold module that is highlighted with a yellow dashed rectangle at the second edge node in Figure 1.

**FL-based LSTM prediction module**: In this module, the LSTM model is trained on multiple Closed-Circuit Television (CCTV) cameras using the FL approach to predict future object occurrences by safeguarding data privacy and security, which were learned from the earlier time-series patterns during the training process. Then, the LSTM module transmits the predicted values to the DQN-based controlling threshold module.

**DQN-based controlling threshold module**: The DQN-based controlling threshold module acts as the decision-making center that intelligently determines the controlling threshold time value in the overall system. Here, the threshold time value represents a timeout for deciding whether to hold or release the DL model. The DQN model receives the predicted object occurrence outcomes generated by the LSTM model. These prediction values are then used as state observations for the DQN model to make a crucial decision (i.e., whether to release or hold the DL model into action). Using Algorithm 1, the threshold time value is continually updated based on the DQN model’s decision. This algorithm operates by continuously monitoring the DQN’s action, which adjusts the threshold time value to determine whether to hold or release the DL model into action. If the DQN’s action suggests holding the model, the motion-tracking threshold is incrementally increased by one second, ensuring a cautious approach. Conversely, if the action indicates releasing the model, the threshold is decreased by one second, facilitating quicker response times to detected events. This iterative process ensures that the threshold adapts intelligently to the system’s needs, optimizing its performance in real time.
**Algorithm 1** Updating threshold time value (θm).**Require:** DQN’s action (DQNact): release = 0, hold = 1, Motion-tracking threshold θm,1:**while** *True* **do**2:    DQN’s action (DQNact)3:    **if** DQNact == 1 **then**4:        Motion-tracking threshold is increased by 1 sec (θm←θm + Δ, θmax)5:    **else**6:        Motion-tracking threshold is decreased by 1 sec (θm←θm - Δ, θmin)7:    **end if**8:**end while**

Algorithm 2 presents the mechanism of the motion-tracking module at the second edge node, which relies on both queue and threshold time values. Algorithm 2 continuously receives frames and object detection data, updating the threshold time value from Algorithm 1 based on the decision of the DQN model. The key decision-making point lies in comparing the duration of an empty queue with the threshold time value: if the queue remains empty for a duration equal to or exceeding the threshold, a command is issued to halt the motion-tracking module, ensuring efficient resource utilization. Otherwise, frames are forwarded to the motion-tracking process, maintaining the system’s responsiveness to detected events.
**Algorithm 2** Controlling threshold module.**Require:** Threshold for motion tracking $\theta_{m}$, DQN’s action ($DQN_{act}$), empty queue $t_{empty}$1:**while** *True* **do**2:    Put received frames and detected number of objects into the queue3:    DQN’s action ($DQN_{act}$) based on LSTM module predictions4:    Update $\theta_{m}$ based upon the DQN’s action5:    **if** $t_{empty}$≥$\theta_{m}$ **then**6:        Transmit a command to the motion-tracking module to stop, accompanied by an activation signal.7:    **else**8:        Transmit the acquired frames to the motion-tracking process via queue mechanisms.9:    **end if**10:**end while**

### 3.1. FL-Based LSTM Prediction Module

FL is used in various applications, including personalized recommendations [35], financial transactions [36], healthcare data analysis [37,38,39], and mobile keyboard predictions [40]. In video surveillance systems, FL offers significant advantages [41]. Firstly, it preserves privacy by keeping raw video data localized, reducing the risk of data breaches. Secondly, it optimizes resource usage by distributing the training process among clients, allowing scalability and efficient utilization of resources. Adopting FL in video surveillance addresses privacy concerns and resource limitations, enabling collaborative learning and distributed data utilization.

Similarly, in the case of the proposed cost-effective video surveillance system, FL enables the LSTM model to be trained on data from multiple cameras without transferring the data to a centralized server, as shown in Figure 2. Every client creates a local model by optimizing its individual objective function, which is then shared with the FL server. Once the FL server collects local models from all participating FL clients, it aggregates them to update a global model. This global model is initially shared with all network clients at the beginning of FL training. Subsequently, the updated global model is distributed to all FL clients, and each FL client refines their local model by incorporating knowledge from the global model. This process of interaction between FL clients and the server continues until the global model achieves the desired level of accuracy, ultimately reaching the target convergence of the model. As a result, FL ensures privacy preservation and mitigates the risk of security breaches. Furthermore, FL enables the LSTM model to be trained in a distributed manner, reducing the strain on processing power and storage capacity. This approach allows for the utilization of a larger and more diverse dataset.

### 3.2. DQN-Based Controlling Threshold Module

The DQN model offers adaptive decision-making capabilities, learning from experience and adjusting its decision-making process based on real-world performance [42]. In our research, we implement the DQN to optimize the threshold time for releasing the DL model in the video surveillance system, as shown in Figure 3.

**Model-Free Approach:** Our approach is model-free, meaning that we do not have access to a predefined mathematical model of the system dynamics. Instead, we utilize the predictions of the LSTM model, which captures the temporal dependencies of object occurrence patterns, as input to the DQN model. The DQN model then learns the optimal threshold time for releasing the DL model based on the LSTM predictions.

**Policy-Based Model:** The DQN model operates based on a policy-based approach, where it learns a policy that maximizes the expected future reward. In our case, the current state of the system is represented by the LSTM predictions. The DQN model takes this state as input and selects actions from the binary action space, deciding whether or not to release the DL model.

By considering various factors such as object appearance patterns, system performance, and resource usage, a DQN model can make intelligent decisions regarding when to trigger the release of the DL model, ensuring efficient resource utilization and improving the overall effectiveness of the smart video surveillance management system. In the following, we highlight the state space, action space, environment space, and reward function for our DQN agent:State Space: The state space refers to the range of possible LSTM predictions, which represent the predicted number of people in the CCTV video per second. This is the information the DQN agent uses to make decisions, helping balance the trade-off between GPU resource conservation and DL model reloading latency.Action Space: The action space defines the two available actions for the DQN agent: either releasing the DL model or holding GPU resources. The agent chooses from these actions to optimize resource allocation based on the state.Environment Space: The environment is the system in which our DQN agent interacts, including the LSTM predictions and GPU resources. The DQN agent interacts with this environment to determine when to release the DL model.Reward Function: The reward function quantifies the desirability of the DQN agent’s decisions. Specifically, a negative penalty (−1) is incurred when the DL model is released and objects appear shortly after. This discourages releasing the model when objects are likely to appear. Conversely, a positive reward (+1) is earned when the DL model is released but no objects appear within a few seconds. This rewards correct decisions to withhold the model when unnecessary. A positive reward (+1) is also earned when the DL model is not released and objects appear within a few seconds. This encourages resource conservation by withholding the model when objects are unlikely to appear. Conversely, a negative penalty (−1) is incurred when the DL model is not released but objects do not appear within a few seconds. This penalizes incorrect decisions to retain the model in such cases.

The training procedure of the DQN-based decision-making model for DL model release consists of several critical components, as shown in Algorithm 3. Replay memory (*D*) stores past experiences for learning, with a set capacity (*N*). The Q-network (*Q*) approximates Q-values, representing expected future rewards for actions in given states. The target network ($Q_{\mathrm{target}}$) mirrors *Q* and initially stabilizes training. The state (*s*) represents the LSTM predictions, serving as input. Various hyperparameters, including the learning rate, discount factor (γ), and exploration strategy, must be configured. The model operates in episodic loops, with episodes defining scenarios for deciding DL model release. Each episode has step loops, where the agent selects actions based on Q-values. Learning involves updating the Q-network with the observed rewards and transitions, while the target network is periodically updated to ensure training stability. The DL model reload check condition influences when to initiate new episodes, balancing resource-saving and model reloading latency avoidance.
**Algorithm 3** DQN training procedure.1:Initialize replay memory *D* with capacity *N*2:Initialize Q network with random weights *Q*3:Initialize target network with weights $Q_{\mathrm{target}}=Q$4:Initialize state *s* (LSTM predictions)5:**for** episode in range(total episodes) **do**6:    Initialize the DL model7:    Set the initial state *s* from LSTM predictions8:    **for** t in range(max_time_steps) **do**9:        Choose action $a_{t}$10:        Execute action $a_{t}$, release DL model if $a_{t}$ is 1, otherwise hold resources11:        Observe the next state $s^{\prime }$ from updated LSTM predictions12:        Observe the reward $r_{t}$ (based on the reward function)13:        Store transition $\left(s_{t},a_{t},r_{t},s^{\prime }\right)$ in *D*14:        Sample random minibatch of transitions $\left(s_{i},a_{i},r_{i},s_{i}^{\prime }\right)$ from *D*15:        Calculate the target Q-values: $Q_{\mathrm{target}}\left(s_{i},a_{i}\right)=r_{i}+\gamma \max_{a}Q\left(s_{i}^{\prime },a\right)$16:        Update the Q-network using a loss function: $L=\frac{1}{2}{\left(Q\left(s_{i},a_{i}\right)-Q_{\mathrm{target}}\left(s_{i},a_{i}\right)\right)}^{2}$17:        Update *Q* using backpropagation and gradient descent18:        Update the target network19:        Set *s* to $s^{\prime }$ for the next iteration20:        **if** DL model reloading latency is over **then**21:           Return to step 6 (start a new episode)22:        **end if**23:    **end for**24:**end for**

## 4. Implementation

We provide the implementation details for building a cost-effective video surveillance management framework. In our testing outline, we examine the hierarchy of two edge nodes for a real-time surveillance system: the first edge node and the second edge node. These edge nodes are interconnected. We selected the Jetson Nano for the first edge node due to its sufficient computing resources for running the detection models. Conversely, we used a desktop equipped with a GeForce RTX 2080 SUPER GPU for the second edge node because it offers more powerful computational capabilities suitable for the motion-tracking models.

The first edge node is responsible for communication with the second edge node and object detection using the YOLO algorithm. Specifically, it utilizes the YOLOv8s [43], a compact model designed for mobile and edge devices with limited resources. To facilitate this communication, the first edge node transmits video frames along with detection information to the second edge node through Python sockets.

The second edge node has multiple functionalities, including predicting future object occurrences (i.e., FL-based LSTM model), DQN-based management of the threshold time value to optimize GPU memory usage, implementing the motion-tracking module, and establishing a connection with the first edge node. The threshold management module adjusts the threshold value, temporarily pausing processes to save GPU memory resources. The Python Thread class manages data flow through the second edge coming from the first edge node. In contrast, the Python Process class is employed to implement the motion-tracking module, and it halts the motion-tracking module in unnecessary situations based on the controlling module to conserve GPU memory at the second edge.

The workflow of the second edge node begins by receiving video frames with detection information through the Python socket from the first edge node. These frames and the detection information are then placed in a queue to enable simultaneous communication through multiple processes, such as the FL-based LSTM module for future object occurrence prediction, DQN-based controlling threshold time, and motion-tracking algorithms. Since the frames with their detection information are placed in a queue, the FL-based LSTM module processes the detection information while the motion-tracking module simultaneously analyzes the frames. The DQN-based controlling threshold module sets and updates the threshold time value based on the LSTM predictions, as described in Algorithm 1. After setting the threshold value, if the queue remains empty for a specified duration by exceeding the threshold time value, a trigger signal with a stop instruction is transmitted to the motion-tracking module, terminating the motion-tracking process, as mentioned in Section 3.

In our experiments, our LSTM model is trained using FL on multiple CCTV cameras. We opted for the RMSE metric to evaluate the accuracy of the DL model’s predictions (i.e., LSTM model), as it allowed us to impose penalties for larger errors [44]. This is necessary because we occasionally encounter significant prediction errors due to unexpected object appearances in video surveillance services. For motion tracking itself, TF pose estimation is used (i.e., Tensorflow-based human pose estimation model).

## 5. Performance Evaluation

In this section, we assess the effectiveness and efficiency of our proposed system. We conduct experiments and analyses to measure key performance metrics, such as object occurrence prediction accuracy, GPU memory utilization, and system responsiveness in terms of performance optimization and resource management.

### 5.1. Evaluation Metrics

We also conduct performance assessments by comparing measurements with prior research to demonstrate the feasibility of our proposed system. Here is a brief overview of the benchmark, previous studies, and intended framework used in the evaluation section:**Baseline**: The baseline system, based on the proposed framework without the FL-LSTM and DQN modules, uses object recognition and motion tracking for accurate evaluation. It keeps GPU memory for motion tracking even when no object is detected in the video frame.**AdaMM**: All proposed modules (i.e., frame differences and management of adaptive processes) are included in AdaMM [11].**CogVSM**: This includes all the previous modules from more advanced models, such as the LSTM prediction and controlling threshold modules in CogVSM [12].**Proposed work**: This covers all proposed modules, such as the FL-based LSTM prediction and DQN-based enhanced intelligent controlling threshold modules.**Proposed work with CNN**: This covers every module in the proposed framework, with the exception of the LSTM model. This framework utilizes a DQN-based enhanced intelligent controlling threshold module and an FL-based convolutional neural network (CNN) model in place of the FL-based LSTM module.

Moreover, we evaluate only one crucial performance indicator obtained from the second edge node, the GPU memory utilization, denoted as $GPU_{m}$[11,12]. The range of $GPU_{m}$ is [0%, 100%]. Table 2 summarizes the parameters and settings of the proposed framework.

About 3600 megabytes (MiB) (total memory of 7979 Mib) are needed to load the model for motion tracking; hence, $GPU_{m}$ is roughly 46%. Moreover, $\theta_{m}$ is the threshold time value. For the evaluation, we use different $\theta_{m}$ values (e.g., $\theta_{m}$ = 10 s and 30 s) for only the AdaMM framework since it uses constant threshold values for releasing DL models.

### 5.2. Model Selection

In this subsection, we select the most accurate time-series prediction model. For this, we experimented with common deep learning architectures designed for time-series prediction problems. By using the RMSE metric as our accuracy measure, we can penalize large errors caused by sudden object occurrences in surveillance videos [44,45]. We tested LSTM, CNN, Gated Recurrent Unit (GRU), and vanilla RNN [46,47] on the dataset described in Table 3. Table 4 illustrates the training results of the aforementioned mainstream DL models, measured using the RMSE metric. Our results show that the LSTM model outperformed other commonly used deep learning models, achieving an RMSE loss accuracy of 0.8046.

The dataset in Table 3 combines people detection information from two distinct sets of video streams. One set, representing urban environments, was sourced from the Shinjuku Kabukicho live camera [48] in Tokyo, Japan, while the other was obtained from the Koh Samui live camera [49] in Laramie, Albany County, USA, capturing rural surroundings. The dataset includes counts of people detected in CCTV videos using the YOLO algorithm every second over a full day from each live camera. The dataset consists of two attributes: time and the number of persons detected. With 172,000 instances, the dataset provides a thorough representation of people’s patterns over two days. The lack of missing values ensures the dataset’s integrity, which allows for more robust analysis. The numerical, integer structure of the attributes emphasizes the exact measurement of people’s presence, allowing for extensive analysis and modeling of time-series patterns.

### 5.3. Evaluation Results of FL-Based LSTM Module

In this subsection, we simulate the training process of our LSTM model using FL on the dataset in Table 3. Figure 4 shows an overall comparison of the convergence of the LSTM model during the training process in terms of federated and centralized learning on the same dataset in Table 3.

We trained the LSTM model for 200 rounds and evaluated the training results using the RMSE metric. Specifically, we first recorded the required communication rounds for both the federated and centralized training results. We also compared the convergence of the LSTM model across different numbers of clients (K) (e.g., K = 1 means centralized training) in a federated way, as shown in Figure 5. We assessed the accuracy of our LSTM model in terms of the RMSE. The convergence results show that the centralized-based LSTM model achieved the lowest 0.79 RMSE value, following the FL-based LSTM models with an increasing number of clients. One significant reason for this is that the dataset was distributed among clients in the FL-based LSTM during the training process, which influenced model performance. While centralized training can produce better outcomes in these circumstances, it is crucial to acknowledge the unparalleled advantages of FL-based LSTM training regarding privacy and security. Moreover, FL enables collaboration among parties while safeguarding data integrity. This makes it an ideal choice where maintaining ownership and data privacy are important. Considering these advantages, our research recommends adopting FL-based LSTM training as the approach for privacy, even though there were some performance differences compared to centralized training, as observed in Figure 4 and Figure 5.

Additionally, we created one example video to demonstrate the prediction accuracy of the LSTM model in video surveillance services. The video’s duration, frame rate, and resolution values were about 300 s, 30 fps, and 1280 × 720, respectively. Figure 6 shows the presence of objects and the forecast of object occurrences. In Figure 6, values of one and zero indicate that the object was either detected or undetected utilizing our YOLO algorithm at the first edge node. In the video, the object was only detected during the intervals [0 s, 70 s], [94 s, 145 s], [152 s, 190 s], and [261 s, 307 s]. However, there was no object in other intervals because the object was not detected.

### 5.4. Evaluation Results of DQN

This subsection demonstrates the advantage of the designed DQN-based controlling threshold module in the energy-efficient video surveillance system. To train our DQN model, we utilized a sequence of LSTM predictions as input states.

Figure 7 shows the average cumulative reward for each episode during the training of the DQN-based algorithm. The DQN was trained for 200 episodes. From Figure 7, we can see that the reward grew rapidly in the initial 20 episodes as it learned quickly to balance the trade-off between saving GPU memory and model reloading latency by learning through trial and error. This is because our reward function was explicitly designed to balance this trade-off. After about 50 episodes, the reward fluctuated slightly at around 25. This shows that our training converged, verifying the efficacy of the designed DQN-based model.

Moreover, we compared the efficacy of our DQN-based controlling module and the EWMA-based controlling module, which was proposed in [12], using a sample video. The comparison shows the controlling threshold methods by statistical technique and deep reinforcement model, as shown in Table 5. The count values represent the detected number of people obtained from the first edge node. The forecast values are the predictions generated by our LSTM model.

The LSTM model predicts the occurrence of an object at $Time_{t+1}$ using the information from $Time_{t}$. For example, at $Time_{10}\left(s\right)$, if the count is 2 and the forecast is 2.623234, it means that the forecasted value at $Time_{10}\left(s\right)$ is a prediction of the object occurrence at $Time_{11}\left(s\right)$, which corresponds to a count of 3.

The $\theta_{m}$(EWMA) and $\theta_{m}$(DQN) values represent the EWMA-based controlling threshold values and DQN-based threshold values, respectively. For our experiments, we selected the threshold time value from within the range of 0 to 2.

In Table 5, we compare the EWMA-based controlling threshold module in [12] and the DQN-based controlling threshold module in our proposed framework by utilizing the LSTM model prediction using the same sample video. The video duration was [0 s, 30 s], denoted as *Time(s)*. Here, *Count* refers to the detected number of people and *Forecast* represents the LSTM prediction based on the sample video. From Table 5, it can be clearly seen that the DQN-based controlling module’s threshold time value was more sensitive and faster in anticipating object absence (i.e., in the intervals [2 s, 5 s], [12 s, 15 s], and [20 s, 27 s] in the sample video) in a video surveillance system compared to the EWMA-based controlling threshold module. This is because our DQN-based controlling threshold module made more accurate decisions compared to the EWMA-based controlling threshold module. This ensures the avoidance of model reloading latency (i.e., 3 s) by balancing the trade-off between saving GPU memory and model reloading latency.

### 5.5. Performance Comparison

This subsection delves into assessing our novel approach by comparing it against prior methodologies. We experiment with real-time video surveillance, including first and second edges, as mentioned in Section 3, using a sample video from a CCTV camera [49].

Moreover, we believe that our object detection model detects people accurately because we followed the approach in [50] to improve the accuracy of people detection in video surveillance systems. In addition, we optimized our YOLO algorithm [43] by employing NVIDIA’s TensorRT [51] for inference purposes, resulting in a twofold increase in processing speed while maintaining accuracy at a level of negligible reduction [52]. Regarding the acceptable precision of our object detection method and accurate prediction of object occurrence patterns, it stands to reason that our proposed technique has the potential to optimize resource consumption, reducing the possibility of model release oversights during object presence.

In the sample video, the object was detected only during the intervals [0 s, 63 s], [97 s, 145 s], [153 s, 185 s], and [266 s, 304 s]. Figure 8 illustrates the performance contrast among five different frameworks: baseline, AdaMM, CogVSM, our proposed framework, and our proposed framework with the CNN model instead of the LSTM model. For a detailed evaluation, we applied constant $\theta_{m}$ values (i.e., $\theta_{m}$ = 10 s and $\theta_{m}$ = 30 s) to the AdaMM framework.

Commencing at time 0 s, all frameworks loaded modules to address the video request. Notably, our proposed framework exhibited the most efficient utilization of GPU memory, followed by our framework with CNN, CogVSM, AdaMM, and lastly the baseline for both $\theta_{m}$ values. This efficiency can be attributed to our approach of releasing GPU memory for motion tracking, guided by the predictions of the LSTM model and the intelligent threshold control facilitated by the DQN model.

Our proposed framework efficiently managed GPU memory at the second edge node by employing predictive object occurrence and adeptly controlling the forecast outcomes. In contrast, our framework with the CNN model followed suit by releasing the DL model because of its low prediction accuracy, followed by AdaMM, which resulted in increased GPU memory consumption because it waited for frames until $\theta_{m}$ seconds, with memory usage varying according to $\theta_{m}$ values, as evident in Figure 8. This was followed by CogVSM, which exhibited intermediate GPU memory usage compared to our approach and AdaMM. Finally, the baseline could not save GPU memory regardless of object absence in the sample video because it always loaded GPU memory regardless of the object absence setting.

When $\theta_{m}$ was set to 10 s for AdaMM, as shown in Figure 8a, our proposed framework with CNN and CogVSM terminated the motion-tracking process three times, each time effectively releasing GPU memory, whereas AdaMM released GPU memory twice. This is because there were intervals at around [145 s, 153 s] where the controlling module in AdaMM did not release the DL model due to the constant $\theta_{m}$ value. Furthermore, when $\theta_{m}$ was set to 30 s for AdaMM, as shown in Figure 8b, AdaMM only freed GPU memory once because of its 30 s waiting setting, whereas our proposed framework, our proposed framework with CNN, and CogVSM released the DL model three times. However, the baseline framework could not release the model even once because it always loaded GPU memory regardless of the object absence setting.

Using the results in Figure 8, we analyzed average GPU memory usage, as illustrated in Figure 9. Notably, when $\theta_{m}$ was set to 10 s, our proposed framework and our proposed framework with CNN demonstrated significantly optimized memory utilization, with rates of 29.23% and 30.11%, respectively. This is because both of them utilized our novel approach incorporating both FL-based LSTM and DQN-based intelligent controlling threshold modules. By utilizing the LSTM model and statistical EWMA technique, CogVSM exhibited competitive performance, with GPU memory consumption at 31.43%, followed closely by AdaMM at 34.98%, employing the constant controlling threshold method. Lastly, the baseline approach exhibited the highest GPU memory usage, at 46.09% because it always loaded GPU memory regardless of object absence in video frames. Similarly, at $\theta_{m}=30$ seconds, the GPU memory usage of the AdaMM framework increased by about 5% from 34.98% to 39.98% because AdaMM released three times with $\theta_{m}=10$ and two times with $\theta_{m}=30$. The GPU memory usage of the remaining frameworks with $\theta_{m}=30s$ remained unchanged.

## 6. Conclusions

In this study, we have presented a comprehensive approach for a cost-effective video surveillance system leveraging edge computing. Our work showcases the potential advantages of combining multiple advanced technologies, specifically the YOLO algorithm for object detection, FL for decentralized training of an LSTM model, and a DQN to control the threshold time value to halt the motion-tracking module in unnecessary cases. Comparative analysis with prior methods, such as a baseline, AdaMM, CogVSM, and our proposed framework with the CNN model instead of the LSTM model, has established the superiority of our proposed framework in terms of GPU memory utilization. Notably, our DQN model demonstrated improved adaptability in threshold value over conventional methods like EWMA. While our results are promising, future research could delve deeper into further optimizing the motion-tracking models, integrating more advanced neural network architectures, or exploring other edge computing paradigms. Additionally, as the surveillance industry evolves, there is a growing need to incorporate more advanced features like anomaly detection, which could be integrated into our system.

## Figures and Tables

**Figure 1 sensors-24-02158-f001:**
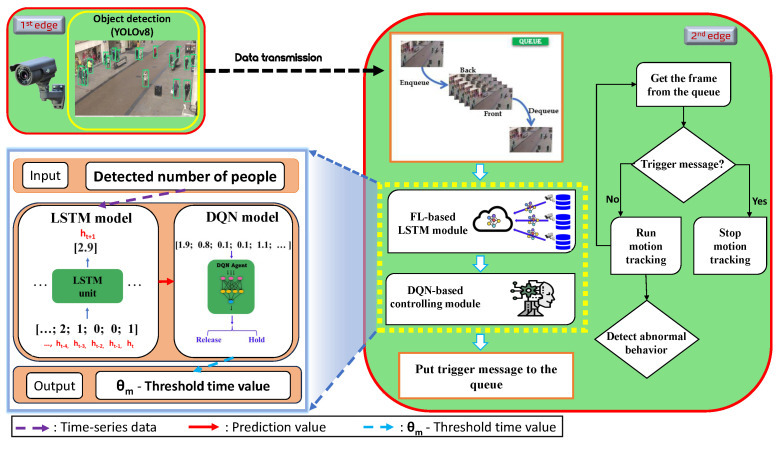
Overall architecture of the proposed framework.

**Figure 2 sensors-24-02158-f002:**
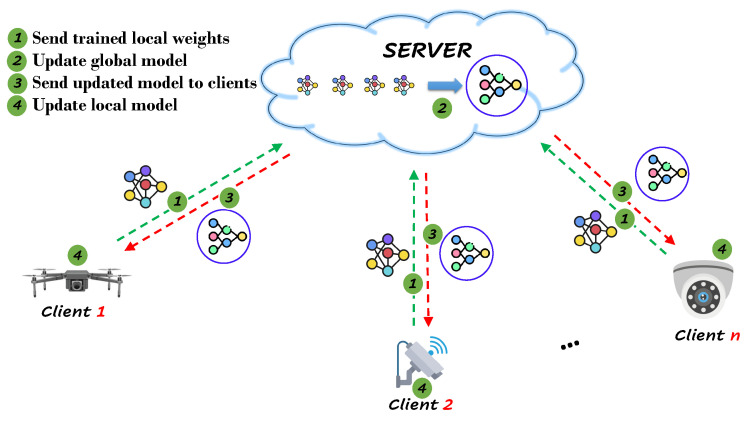
FL in video surveillance.

**Figure 3 sensors-24-02158-f003:**
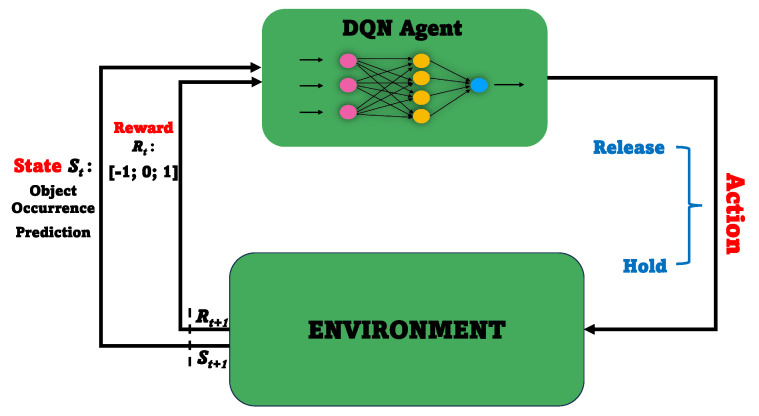
DQN in video surveillance.

**Figure 4 sensors-24-02158-f004:**
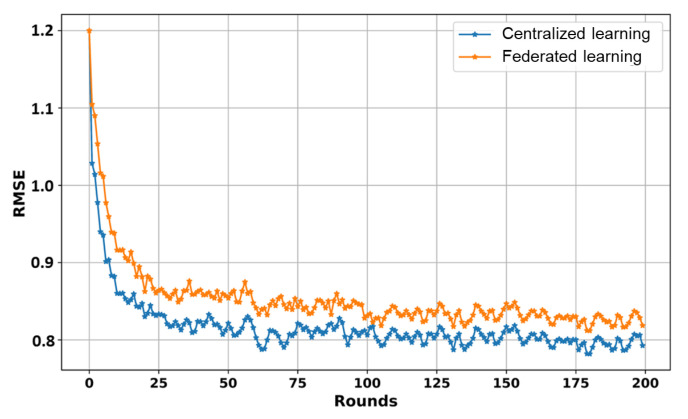
Comparison of convergence results in centralized and federated learning.

**Figure 5 sensors-24-02158-f005:**
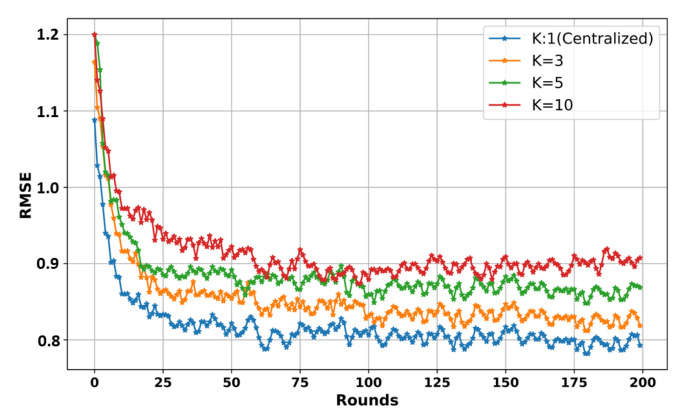
Comparison test results on different K clients.

**Figure 6 sensors-24-02158-f006:**
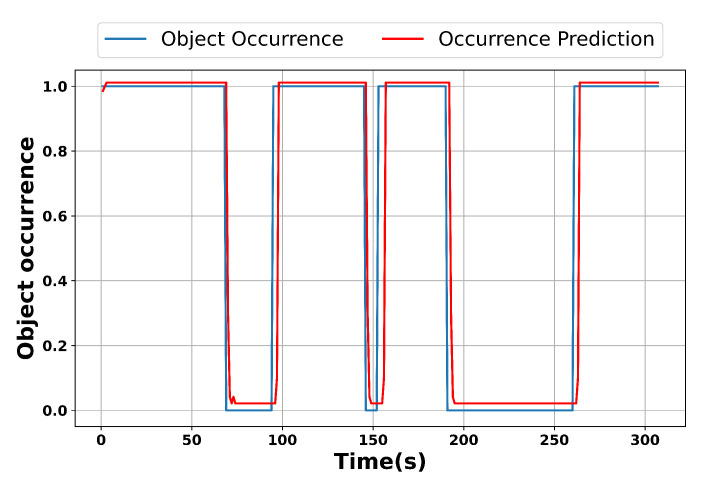
Object occurrence and prediction in the sample video.

**Figure 7 sensors-24-02158-f007:**
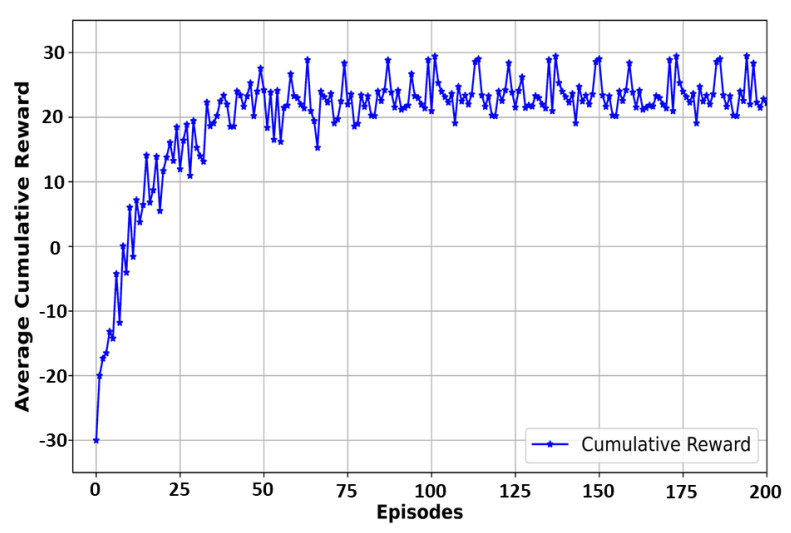
Convergence of the DQN model during training.

**Figure 8 sensors-24-02158-f008:**
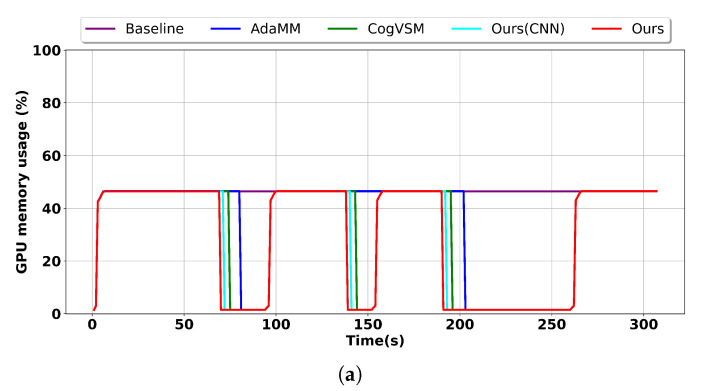
Comparison of the GPU memory usage on the second edge node for our proposed framework, our proposed framework with CNN, CogVSM, AdaMM, and the baseline. (**a**) $\theta_{m}$ = 10 s. (**b**) $\theta_{m}$ = 30 s.

**Figure 9 sensors-24-02158-f009:**
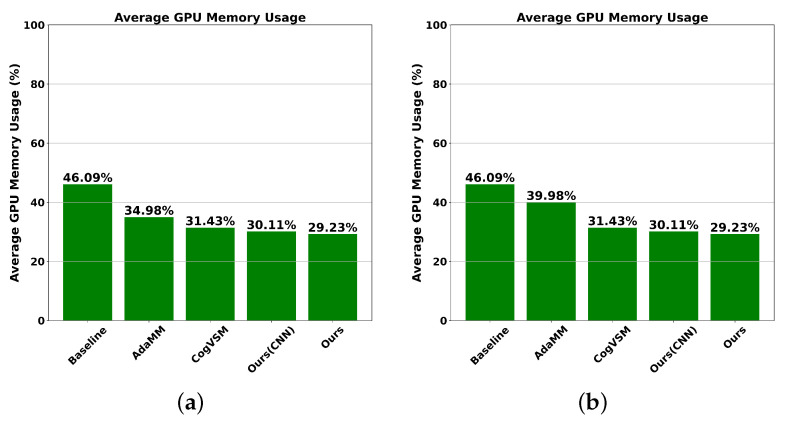
Comparison of average GPU memory usage on the second edge node for our proposed framework, our proposed framework with CNN, CogVSM, AdaMM, and the baseline. (**a**) $\theta_{m}=10$ s. (**b**) $\theta_{m}=30$ s.

**Table 1 sensors-24-02158-t001:** Summary of related works on video surveillance management systems.

Publication	Year	Aim of Research	Proposed Solution
Alam et al. [23]	2019	Cost-effective abnormal event detection	Focused on reducing delays and network data usage.
Lee et al. [24]	2019	Cost-effective precise object tracking	Suggested TLD (Tracking, Learning, and Detecting) approach.
Xu et al. [19]	2021	Cost-effective video surveillance system	Suggested FL-YOLO algorithm for real-time video analysis algorithm.
Rajavel et al. [20]	2022	Cost-effective video surveillance system	Leveraged edge computing and optimization mechanisms.
Farahdel et al. [25]	2022	Cost-effective abnormal event detection	Suggested an efficient video transmission algorithm.
Naveen et al. [22]	2022	Saving GPU resources	Suggested reducing the number of non-contributing parameters.
Kim et al. [11]	2021	Saving GPU resources	Introduced a constant threshold for DL model release.
Bazarov et al. [12]	2023	Saving GPU resources	Applied LSTM and EWMA for adaptive DL model release.

**Table 2 sensors-24-02158-t002:** Configuration and hyperparameters of the second edge node.

Parameter	Value
GPU memory usage, $GPU_{m}$ (%)	[0%, 100%]
Threshold for stopping the process, $\theta_{m}$ (s)	[10 s, 30 s]

**Table 3 sensors-24-02158-t003:** Dataset description.

Parameter	Value
Characteristics	Bivariate,Time−Series
Number of attributes	2,(time,number of people)
Attribute characteristics	Numerical,Integer
Missing values	No
Number of instances	172,000

**Table 4 sensors-24-02158-t004:** Training results of mainstream DL models using the RMSE metric.

Model	RMSE
Long Short-Term Memory (LSTM)	0.8046
Convolutional Neural Network (CNN)	0.8101
Gated Recurrent Unit (GRU)	0.8198
Vanilla Recurrent Neural Network (RNN)	0.8403

**Table 5 sensors-24-02158-t005:** Comparison of DQN and EWMA.

Time(s)	Count	Forecast	$\theta_{m}$ (EWMA)	$\theta_{m}$ (DQN)
1	2	1.248076	2	2
2	1	0.710065	2	1
3	0	0.363475	1	2
4	0	0.891768	2	2
5	1	2.601207	2	2
6	3	2.819257	2	2
7	3	1.499124	2	2
8	1	0.512877	2	2
9	0	1.804097	2	2
10	2	2.623234	2	2
11	3	1.567843	2	2
12	1	0.623472	2	1
13	0	0.432745	2	2
14	0	0.254022	1	2
15	0	0.812373	2	2
16	1	1.675492	2	2
17	2	3.604922	2	2
18	4	2.214837	2	2
19	2	1.327493	2	2
20	1	0.654289	2	1
21	0	0.456283	2	0
22	0	0.249513	1	0
23	0	0.148536	0	0
24	0	0.031254	0	0
25	0	0.031254	0	0
26	0	0.031254	0	0
27	0	0.031254	0	0
28	0	0.885301	1	1
29	1	2.012847	2	2
30	2	2.003821	2	2

## Data Availability

Data are contained within the article.

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
