# Peer review of "Deep Reinforcement Learning-Empowered Cost-Effective Federated Video Surveillance Management Framework"

_sensors, 2024, doi:10.3390/s24072158_

Round 1

Reviewer 1 Report

Comments and Suggestions for Authors

The article focuses on the issue of computing and memory resource consumption in video surveillance systems, which is a practical and challenging topic. With the increasing application of video surveillance systems in various fields, how to improve their efficiency and reduce costs has become an important issue.

The article proposes a federated video surveillance management framework based on deep reinforcement learning, which is a novel approach. Through a two-level edge computing structure, the framework realizes object detection and improves system efficiency, showing its innovation and practicality.

The article conducted detailed experiments and analysis on the proposed framework, including evaluations of its performance, efficiency, and cost. This provides readers with an opportunity to gain a deeper understanding and evaluate the framework.

The article has a clear structure, rigorous logic, fluent language, and is easy to understand. Each section has been fully elaborated and explained, allowing readers to easily follow the author's ideas.

However, there are still the following shortcomings, and I believe that if the author can further revise it, it will make the article more perfect.

 In certain parts of the article, such as "primary edge undertakings object detection" and "two tiered edge computing structures", detailed explanations or background information are not provided, which may confuse readers.

 Although the article has conducted detailed experiments and analysis on the proposed framework, there is a lack of comparison with other relevant methods or frameworks. This comparison can help readers better understand the advantages and disadvantages of the framework.

 Although the article mentions experimental data and results, no specific charts or data tables are provided to visually display these data. In addition, there is insufficient analysis and discussion of experimental data, and the underlying meanings behind these data have not been fully explored.

Reviewer 2 Report

Comments and Suggestions for Authors

Comments on the Quality of English Language

Round 2

Reviewer 1 Report

Comments and Suggestions for Authors

After the authors' revision, the quality of the article has been improved to some extent and basically meets the requirements for publication. It is recommended that the authors replace the pictures in the experimental part with high-resolution figures.

Reviewer 2 Report

Comments and Suggestions for Authors

I would like to thank the authors effort in answering all the presented questions.

There is still need for some improvements, however it can be accepted in current form.

Congratulations.